# Artificial Diet Assay Screening of Candidate RNAi Effectors Against *Myzus persicae* (Hemiptera)

**DOI:** 10.3390/insects16111086

**Published:** 2025-10-23

**Authors:** Amol Bharat Ghodke, Stephen J. Fletcher, Ritesh G. Jain, Narelle Manzie, Neena Mitter, Karl E. Robinson

**Affiliations:** Queensland Alliance for Agriculture and Food Innovation, Centre for Horticultural Sciences, The University of Queensland, St. Lucia, QLD 4072, Australia; amol.ghodke@csiro.au (A.B.G.); s.fletcher@uq.edu.au (S.J.F.); ritesh.jain@southernrna.com.au (R.G.J.); n.manzie@uq.edu.au (N.M.); n.mitter@uq.edu.au (N.M.)

**Keywords:** RNA interference, aphids, crop protection

## Abstract

**Simple Summary:**

The green peach aphid (*Myzus persicae*) is one of the most damaging insect pests affecting vegetable and ornamental crops worldwide, due to its feeding activity and ability to transmit plant viruses. Although synthetic chemical pesticides are commonly used to manage *M. persicae*, their widespread use raises concerns about human health, environmental contamination, pesticide residues in food, harm to beneficial insects, and the development of resistance. As a result, there is growing interest in safer, eco-friendly alternatives. In this study, we used sucrose-based artificial diet assays supplemented with double-stranded RNA (dsRNA) to identify RNA interference (RNAi) effectors capable of inducing gene silencing and mortality in *M. persicae* through ingestion. Defining effective gene targets is a critical first step toward developing a non-toxic, environmentally sustainable RNAi-based technology for controlling *M. persicae* and the viruses it spreads.

**Abstract:**

Aphids are sap-sucking pests that cause substantial damage to fruit and fibre crops through direct feeding and transmission of plant viruses. While chemical pesticides remain the primary method of control, their use raises concerns related to human health, environmental contamination, pesticide resistance, and impacts on beneficial insects. As a sustainable alternative, spray-on double-stranded RNA (dsRNA) technology offers a promising approach to induce RNA interference (RNAi) in target pests. For RNAi to be effective against sap-sucking insects like the green peach aphid (*Myzus persicae*), it is essential to identify genes whose silencing disrupts vital physiological functions. In this study, artificial diet (AD)-based feeding assays were used to evaluate dsRNAs targeting eight genes involved in neural function, osmoregulation, feeding behaviour, and nucleic acid/protein metabolism. dsRNAs were administered individually, in combinations, or as a multi-target stacked construct. After 98 h of feeding, aphid mortality ranged from 14 to 72% (individual targets), 78–85% (combinations), and 54% (stacked construct). Transcript knockdown varied from 6.3% to ~54%, though a consistent correlation with mortality was not always observed. The gene targets and combinatorial dsRNA strategies identified in this study provide a foundation for developing RNAi-based crop protection technologies against *M. persicae* infestation.

## 1. Introduction

Aphids, belonging to the order *Hemiptera*, are significant agricultural pests due to their ability to transmit plant viruses and cause direct feeding damage. Managing aphid populations is challenging and has traditionally relied on synthetic chemical insecticides. However, increasing resistance among aphid populations [1,2], growing awareness of the harmful effects of insecticides on pollinators and beneficial insects [3], and concerns about the persistence of ecologically active chemicals in the environment [4,5] have prompted a shift toward more sustainable and environmentally friendly pest control strategies.

One promising alternative is the topical application of double-stranded RNA (dsRNA) to plants to trigger RNA interference (RNAi). RNAi is a sequence-specific, endogenous mechanism involved in antiviral defence and gene regulation at both transcriptional and post-transcriptional levels and is conserved across eukaryotes [6]. As a crop protection tool, RNAi has been successfully employed in transgenic technologies for over three decades [7,8]. However, transgenic approaches face several limitations, including long development timelines, lack of transformation protocols for key crop species, regulatory hurdles, and consumer resistance to genetically modified fruit and fibre crops [9].

In contrast, spray-on RNAi offers a non-transgenic alternative by applying dsRNA directly to plants, thereby inducing transient gene silencing in target pests or pathogens. Although still in early development, spray-on RNAi has shown promise against a range of agricultural threats, including the Colorado potato beetle [10], fungal pathogens [11,12,13,14], and plant viruses [15,16,17]. Studies have demonstrated that applied dsRNA or siRNA can translocate from plants to insects [18], fungi [19], and even from insects to their parasites, such as *Apis mellifera* to *Varroa destructor* [20], supporting the viability of spray-on RNAi as a transient but effective crop protection strategy.

In aphids, RNAi-mediated gene silencing has been achieved through various delivery methods, including injection [21], topical contact [22], ingestion via artificial diet (AD) [23,24], and transgenic plant expression of dsRNA targeting aphid transcripts [25,26,27] (see Appendix A). Additionally, siRNAs generated in transgenic plants have been shown to move from the phloem into aphids via the stylet, triggering gene silencing [26]. Despite these successes, the efficacy of ingested dsRNA remains debated due to potential degradation by nucleases [28,29] and genotypic variability within aphid colonies [30].

To develop an effective spray-on RNAi product, several challenges must be addressed, including dsRNA delivery methods, environmental stability, uptake mechanisms, and processing within plant and insect tissues. Most critically, identifying gene targets that reliably induce mortality upon ingestion is essential. While numerous genes have been explored in transgenic contexts (Appendix A), their suitability for spray-on applications requires further validation.

In this study, we aimed to identify dsRNA effectors that, when ingested by *Myzus persicae*, disrupt metabolic function and induce mortality. Using artificial diet feeding assays, we demonstrate that dsRNAs targeting *Acetylcholinesterase (ACE)*, *Aquaporin (AQP)*, and *RNAHelicase* significantly increase aphid mortality. Moreover, combinations and stacked constructs of selected dsRNAs exhibit additive effects, further enhancing lethality. Consistent with prior research, we observed that transcript knockdown was often limited, despite significant mortality, suggesting complex mechanisms underpinning RNAi efficacy in aphids.

## 2. Materials and Methods

### 2.1. Aphid Culture

*Capsicum annuum* var. ‘Yollo Wonder’ (*C. annuum* (YW)) plants were used for *M. persicae* colony maintenance. The aphid colony was initially propagated from several individual wild-harvested *M. persicae* identified on a common dandelion, *Taraxacum officinale*, in Ipswich, QLD Australia. Collected aphids (n~20) were starved for ~16 h at 4 °C and placed on *C. annuum* seedlings. Plants were monitored for indicative viral infection pathology and assessed by DAS-ELISA (Agdia, Elkhart, IN, USA) against *Potyviridae*, Cucumber mosaic cucomovirus (CMV), Tomato spotted wilt virus (TSWV), and Bean Common Mosaic virus (BCMV) at day 30 post placement. Stock plants for colony maintenance were grown in 150 mm square pots from seed under glasshouse conditions. The *M. persicae* colony was maintained in a climate-controlled plant growth cabinet with a 16:8 h light/dark photoperiod, a day/night temperature of 25/18 °C, and relative humidity of 70% in vented ‘Bug dorm’ (47.5 cm^3^) cages at the Center for Horticultural Sciences laboratory, QAAFI, The University of Queensland, St. Lucia, Brisbane, Australia.

### 2.2. M. persicae Genes Targeted for RNAi Bioassays

Primers for dsRNA template amplification from *M. persicae* transcripts were designed as previously described [31]. Briefly, regions with minimal homology to the human, honeybee, and monarch butterfly transcriptomes were identified, and primers were selected to generate amplicons of approximately 300 bp. Several different criteria were used for target selection, viz, (1) previous successful reports, (2) target gene expression profile, (3) understanding gene family size (if known), and (4) target gene function. The selected target gene at least satisfied two to three of the above criteria. Candidate *M. persicae* genes targeted for RNAi encompassing neural functioning: Acetylcholinesterase (*ACE*) (GenBank accession # KJ561353) and Transcription factor glial cells missing (*Tfglial*) (XM_022324935); Osmoregulation: Sucrose Transporter (*SUC*) (KR047101); Aquaporin (*AQP*) (KR047100); Probing/feeding behaviours: Salivary gene (*C002*) (EC389531.1); or nucleic acid and protein metabolism: ATP-dependent RNA helicase (*RNAHelicase*) (XM_022316612), Proteasome subunit alpha type (*ProtSubAlpha*) (XM_022311085), and S-adenosylmethionine synthase (*S-AdMethSynth*) (XM_022314690) and a stacked construct incorporating 166 bp of *AQP*, *C002*, *ACE* and *SUC* each of *M. persicae* target genes and designated 830, respectively, were synthesised by Genscript (USA) with respective plasmid clones used to amplify linear templates for in vitro dsRNA synthesis. Target regions of respective genes are presented in Appendix A.

### 2.3. In Vitro Transcription of dsRNA Effectors

Linear DNA templates for in vitro dsRNA synthesis were PCR amplified using chimeric primers, as shown in Appendix A. Forward and reverse chimeric primers incorporated a 5’ T7 polymerase promoter sequence followed by ~20 bp of a homologous sequence of the respective cloned target gene. All molecular biology reagents were sourced from Meridian BioSciences, Cincinnati, OH, USA, unless stated otherwise, and manufacturers’ suggested reaction components and conditions followed. Reactions (50 µL) contained 10× reaction buffer, 5U MyTaq polymerase, 0.4 μM of each respective reverse/forward chimeric primer (Appendix A), and 0.2 ng µL^−1^ respective pUC57 plasmids. PCR cycling conditions were initial denaturation at 95 °C for 1 min, followed by 30 cycles of 95 °C for 15 s, 54 °C for 15 s, and 72 °C for 10 s, final extension of 72 °C for 5 min, and a completion hold at 12 °C. Amplicons were resolved on a 0.8% TAE agarose gel for 1 h at 90 V and purified using the Wizard SV gel purification kit (Promega, Madison, WI, USA). Purified linear template (~1.5 µg) was used for in vitro synthesis of dsRNA using the HiScribe^®^ T7 High Yield RNA Kit (New England Biosciences, Ipswich, MA, USA) as per manufacturer’s recommendations.

### 2.4. Aphid Artificial Diet Feeding Bioassay

To assess the potency of in vitro synthesised dsRNA to induce mortality and/or gene knockdown, aphids were fed respective RNAi effectors in an artificial diet (AD) bioassay individually or in selected combinations. Mortality percentages and changes in relative gene target transcript expression were determined by direct aphid counts and RT-qPCR, respectively, from the average of three independent experiments. Briefly, aphids were harvested from detached colony leaves by gentle heating on a desktop lamp; mobile aphids were knocked from the leaf into a container, to which they proceeded to climb the walls of. Approximately 25–30 instar nymphs, 3 to 4, and adult aphids that had accumulated on the rim of the container were harvest by gentle vacuum pooter and deposited into 35 mm Petri-dishes, and enclosed by stretched parafilm. Aphids were maintained on 200 µL of a 30% sucrose-only diet or 30% sucrose diet containing a total of 300 ng µL^−1^ of respective dsRNA controls or RNAi effectors enclosed by a second piece of parafilm, creating the diet sachet. Bioassays where two individual dsRNA effectors were assayed in combination, each dsRNA effector was supplied at 150 ng µL^−1^ for a total dsRNA concentration of 300 ng µL^−1^. Each bioassay was repeated in triplicate, utilising four replicates with the following groups: 30% sucrose, 30% sucrose with non-specific dsRNA control of either Green fluorescent protein dsRNA (GFP-dsRNA) or Cucumber mosaic cucomovirus silencing suppressor-2b dsRNA (CMV 2b-dsRNA). Aphids in each replicate were counted at 24 h intervals up to 98 h post placement, averaged and the percent of mortality was defined by the difference between aphid numbers at the start of the assay and the end of the assay. Primary aphid count was conducted at 24 h to account for any handling/transfer mortality. A subset of aphids from the treatment groups was removed at 72 h to assess target gene knockdown [32].

### 2.5. Quantitative Reverse Transcription PCR (qRT–qPCR)

Primer 3 (V0.4.0) (http://bioinfo.ut.ee/primer3-0.4.0/ (accessed on 9 Ocotber 2019)) was used to design target-specific gene primers for use in one-step RT-qPCR. Primers were designed to amplify a region outside and 5’ to the sequence targeted by the respective dsRNA effectors. Reactions were performed using Rotor gene Q thermocycler (Qiagen, Hilden, Germany) using Rotor gene software (version 1.7.87). Relative target gene expression was quantified from TriSure (Meridian Biosciences) extracted and TURBO DNase I (Invitrogen, Carlsbad, CA, USA) treated total RNA by qRT-PCR with the following components and conditions. Each 20 µL reaction contained: 2 × Sensifast No ROX One-Step Mix, 0.4 µM of respective forward and reverse primer (Appendix A), 0.2 µL reverse transcriptase, 0.4 µL RNase Inhibitor and 10 ng of DNase I-treated template RNA. Cycling conditions included 10 min at 45 °C and 2 min at 95 °C followed by 35 cycles of 95 °C for 5 s, 60 °C for 10 s and 72 °C for 5 s, with melt curve analysis conducted post-amplification. Relative quantitation of fold-difference between control and treatment groups utilised the 2^−ΔΔCt^ method [33] by normalising the expression of *M. persicae* target genes against the housekeeping (HKs) reference gene ribosomal protein L7 (RpL7) [34,35].

### 2.6. Graphs and Statistical Analysis

Graphical outputs and statistical significance between control and dsRNA-supplemented sucrose diet treatments were analysed by One-Way ANOVA using Tukey’s multiple comparison test with single-pooled variance in GraphPad PRISM 8. Significantly different means are denoted using asterisk, * *p* < 0.05, ** *p* < 0.01, *** *p* < 0.005, **** *p* < 0.0001.

## 3. Results

### 3.1. Artificial Diet Bioassays

#### 3.1.1. Aphid Mortality Counts Fed dsRNA Effectors Individually

AD bioassay results are summarised in Table 1 and graphically represented in Figure 1 (Panels A–H, Top) with raw mortality count data presented in Appendix A. Aphids were fed dsRNA effectors targeting genes involved in neural function (*ACE*, *Tfglial*), osmoregulation (*AQP*, *SUC*), feeding behaviour (*C002*), and nucleic acid/protein metabolism (*RNAHelicase*, *ProtSubAlpha*, *S-AdMethSynth*), either individually, in combinations (AQP/ACE, AQP/C002, ACE/C002; see Figure 2A–C, Top), or as a multi-target stacked construct (830-dsRNA; see Figure 3). Significant mortality was observed in aphids fed dsRNAs targeting *AQP*, *C002*, *ACE*, and *RNAHelicase*, compared to sucrose-only and non-specific dsRNA controls (GFP-dsRNA, CMV 2b-dsRNA). Specifically, *AQP*-dsRNA induced 45.8% mortality, while *C002*-dsRNA caused a modest but significant 16.7% mortality. In contrast, aphid numbers increased by 14.5% and 19.5% in the sucrose and GFP-dsRNA control groups, respectively (Figure 1A,B, Top). *ACE*-dsRNA resulted in 56% mortality, compared to 23% (sucrose) and 19% (GFP-dsRNA) (Figure 1C, Top). *RNAHelicase*-dsRNA led to the highest observed mortality at 71.8%, significantly greater than the 11% and 26.6% mortality seen in sucrose and CMV 2b-dsRNA controls (Figure 1D, Top). In contrast, dsRNAs targeting *SUC* and *ProtSubAlpha* did not induce significant mortality, with rates of 25% and 14%, respectively—comparable to control treatments (sucrose: 23–33%; non-specific dsRNA: 14–36%) (Figure 1E,F, Top). Interestingly, aphids fed *Tfglial*-dsRNA and *S-AdMethSynth*-dsRNA showed a net increase in population (+4.4% and +13%, respectively), suggesting these targets may not be suitable for inducing RNAi-mediated mortality (Figure 1G,H, Top).

#### 3.1.2. Relative Gene Expression (RGE)

Relative gene expression was assessed using real-time qRT-PCR to evaluate transcript abundance following ingestion of dsRNA effectors. Target genes included *AQP*, *C002*, *ACE*, *SUC*, *ProtSubAlpha*, *RNAHelicase*, *Tfglial*, and *S-AdMethSynth*, tested either individually, in selected combinations (AQP/ACE, AQP/C002, ACE/C002; see Figure 2A–C, Lower), or as part of the stacked construct (830-dsRNA; see Figure 3). Among neural function targets, *ACE*-dsRNA and *Tfglial*-dsRNA induced significant transcript reductions of 53.7% and 44%, respectively, compared to non-specific dsRNA controls (Figure 1C,G, Lower). For osmoregulatory genes, *AQP* expression increased by 27% following dsRNA ingestion, though this change was not statistically significant and was comparable to the 16% increase observed in GFP-dsRNA controls (Figure 1A, Lower). Similarly, *SUC* expression increased by ~19% relative to GFP-dsRNA controls (Figure 1E, Lower). Genes involved in nucleic acid and protein metabolism showed varied responses. *RNAHelicase* expression increased by 18.3%, similar to the 17.3% increase observed in CMV 2b-dsRNA controls (Figure 1D, Lower). *S-AdMethSynth* expression was reduced by 41% compared to sucrose-only controls. However, no significant difference was observed between *S-AdMethSynth*-dsRNA and CMV 2b-dsRNA treatments, the latter also inducing a 26% knockdown (Figure 1F, Lower). *ProtSubAlpha*-dsRNA resulted in a modest but statistically significant 6.3% reduction in transcript levels compared to both sucrose and CMV 2b-dsRNA controls (Figure 1F, Lower). For the salivary gene *C002*, dsRNA ingestion did not result in significant knockdown. Instead, transcript levels increased modestly by 14% in the *C002*-dsRNA group and by 10% in the GFP-dsRNA control group (Figure 1B, Lower).

#### 3.1.3. dsRNA Effector Combinations

To evaluate potential additive effects of RNAi-induced mortality, *M. persicae* were fed selected combinations of dsRNA effectors targeting *AQP/ACE* (Combination I), *AQP/C002* (Combination II), or *ACE/C002* (Combination III), each supplied at 150 ng/µL^−1^ (total 300 ng/µL). Mortality outcomes were compared to sucrose-only and GFP-dsRNA controls (Figure 2A–C, Top). Combination I (AQP/ACE) resulted in 78.7% mortality, significantly higher than the 11.4% (sucrose) and 37% (GFP-dsRNA) observed in controls. Notably, the GFP-dsRNA group showed an unexpectedly high mortality, and a significant difference between control groups was observed (Figure 2A, Top). Combination II (AQP/C002) induced 85.2% mortality, compared to 14.9% and 22.6% in sucrose and GFP-dsRNA controls, respectively (Figure 2B, Top). Combination III (ACE/C002) led to 79.4% mortality, with control groups showing 24.6% (sucrose) and 16.5% (GFP-dsRNA). No significant differences were observed between control groups in Combinations II and III (Figure 2C, Top). These results suggest that combining compatible dsRNA effectors can enhance RNAi efficacy and aphid mortality, supporting the concept of additive or synergistic effects. RGE analysis revealed variable transcript responses across combinations. In Combination I, *AQP* and *ACE* showed non-significant reductions of 5.5% and 13.5%, respectively, while GFP-dsRNA induced an 18% increase in *ACE* expression (Figure 2A, Lower). In Combination II, both *AQP* and *C002* exhibited significant transcript reductions in ~59.7% and 33.7%, respectively, compared to sucrose and GFP-dsRNA controls (Figure 2B, Lower). In Combination III, *ACE* and *C002* showed non-significant reductions of 35.3% and 36.7%, respectively. GFP-dsRNA treatment resulted in an average 18.7% decrease in gene expression across tested targets (Figure 2C, Lower). These findings indicate that while transcript knockdown is variable, combinations of dsRNA effectors can produce enhanced mortality, likely through multi-pathway disruption rather than direct correlation with gene expression levels.

#### 3.1.4. Stacked Construct

The mortality of *M. persicae* fed on 300 ng µL^−1^ of 830-dsRNA targeting C002, ACE, AQP and SUC, showed a significant 54% reduction in aphid numbers as compared to those fed on sucrose (3.8%) or non-specific-dsRNA (15.9%) (Figure 3A), suggesting a potential additive effect can be induced with a multi-target ‘stacked’ sequence molecule. Relative gene expression of the respective 830-dsRNA target genes (Figure 3B) showed a statistically significant simultaneous knockdown of both ACE and AQP genes by 49% and 24%, respectively, whereas only a moderate, non-statistically significant reduction in C002 gene expression of 17.7% was achieved. Contrastingly, the RGE of SUC showed a 19.3% increase in gene expression when compared to sucrose-only control.

## 4. Discussion

This study evaluated the insecticidal potential of eight dsRNA effectors targeting key physiological genes in *Myzus persicae*, delivered via artificial diet either individually, in combination, or as a multi-target stacked construct. We demonstrated that dsRNAs targeting *ACE*, *AQP*, and *RNAHelicase*, as well as selected combinations and the stacked construct (830-dsRNA), induced significant mortality ranging from ~45% to 72% within three days. These findings support the feasibility of RNAi-based biopesticide strategies for aphid control and highlight the potential of spray-on RNAi technologies for sap-sucking insects.

The *ACE* gene encodes acetylcholinesterase, a critical enzyme in neurotransmission at cholinergic synapses and neuromuscular junctions in both vertebrates and invertebrates [36]. Due to its essential role in neural function, *ACE* has been a frequent target in transgenic RNAi studies, including those using *Nicotiana tabacum* [37]. In our assays, ingestion of *ACE*-dsRNA resulted in 56% mortality and a 54% reduction in transcript abundance, consistent with previous transgenic findings.

Osmoregulation is vital for aphids due to the high osmotic pressure of phloem sap. *AQP* encodes a water-specific membrane channel protein localised to the gut, and its silencing has previously been shown to disrupt haemolymph balance in *A. pisum* and *M. persicae* [27,38]. In our study, ingestion of *AQP*-dsRNA led to 45.8% mortality, aligning with prior findings despite limited transcript knockdown. This discrepancy may reflect population-level genetic diversity, as previous studies have shown variable RNAi susceptibility within aphid colonies [30].

Notably, *RNAHelicase*-dsRNA induced the highest mortality (71.8%), despite a modest increase in transcript abundance. RNA helicases are multifunctional enzymes involved in DNA/RNA unwinding, transcription, translation, and RNA silencing [39]. To our knowledge, this is the first report of *RNAHelicase* as an RNAi target in aphids. The observed mortality, despite increased transcript levels, suggests complex post-transcriptional or tissue-specific effects not captured by whole-animal qRT-PCR.

The lack of correlation between transcript knockdown and mortality has been reported in other RNAi studies [30,40], and may be attributed to tissue-specific gene expression, biological variability, or genotypic differences within aphid populations [41]. Yoon et al. (2020) [30] demonstrated that RNAi susceptibility in pea aphids varies significantly both between and within colonies, with some individuals being completely resistant. Our results likely reflect similar heterogeneity in *M. persicae*, as high mortality was observed even when transcript reductions were minimal.

Furthermore, the efficacy of RNAi application via ingestion in our study has most likely been affected by the presence of gut nucleases [28,29]. Suggesting, for the future, an ‘RNAi of RNAi’ approach by incorporating dsRNA effectors targeting nuclease genes may achieve an additive effect.

Unexpected outcomes were observed for *Tfglial* and *S-AdMethSynth*. Despite significant transcript knockdown (44% and 41%, respectively), aphid populations increased in both treatment groups. *Tfglial*-dsRNA resulted in a 4.4% increase in aphid numbers, while *S-AdMethSynth*-dsRNA led to a 13% increase. These findings suggest that transcript reduction alone may not be sufficient to induce mortality and point to possible functional redundancy or compensatory mechanisms that mitigate the effects of gene silencing. Further investigation is warranted to clarify the biological basis of these disparities.

Other targets, including *SUC*, *C002*, and *ProtSubAlpha*, did not induce significant mortality or gene knockdown compared to controls. However, transgenic delivery of dsRNAs targeting *SUC* and *C002* has previously shown physiological effects, such as increased osmotic pressure and impaired feeding [27].

A key objective of this study was to assess whether combining dsRNA effectors—either as mixtures or stacked constructs—could enhance RNAi efficacy. Our results show that combinations of dsRNAs targeting *AQP*, *ACE*, and *C002* achieved mortality rates of 78–85%, while the stacked construct (830-dsRNA) induced 54% mortality. These effects were achieved using reduced dsRNA concentrations (150 ng/µL per target) and shorter sequence lengths (166 bp), suggesting that multigene targeting can compensate for reduced siRNA pool size and enhance insecticidal potency.

Interestingly, RGE results for *SUC*, *ACE*, and *C002* in the stacked construct were comparable to those observed when the genes were targeted individually. This indicates that the increased mortality observed in combination treatments likely results from cumulative physiological disruption rather than enhanced transcript suppression. These findings support previous reports that multi-target RNAi can yield greater homeostatic disruption than single-gene silencing [27].

Moreover, the variability in mortality across different combinations suggests that there is no known universal rule for selecting compatible gene targets. Empirical testing remains essential to identify effective pairings and optimise RNAi formulations.

In general, mortality rates exceeding 60% are considered exceptional in transgenic RNAi approaches [42]. In this study, we achieved mortality rates above 78% using artificial diet delivery, and 54% with a stacked construct, despite limited transcript knockdown. These results reinforce the potential of combinatorial RNAi strategies to enhance pest control efficacy and support the development of spray-on RNAi technologies for sustainable crop protection.

These findings further validate the strategy of targeting multiple critical genes in *M. persicae* to amplify the RNAi effect, disrupt insect homeostasis more effectively, and increase mortality. By defining effective individual, combinatorial, and stacked dsRNA effectors, this study lays the groundwork for a holistic approach to aphid control.

Building on our previous research into mitigating virus infection in plants using RNAi-based technologies [16,17,31,43], this work contributes to a broader strategy aimed at disrupting the insect–virus–plant trichotomy. By deploying spray-on RNAi effectors that target aphid metabolic processes as well as challenging the ability of a transmitted virus to establish infection in plants, we aim to provide a clean, green, and persistent crop protection system. This dual-action approach offers a sustainable alternative to chemical pesticides, with the potential to reduce aphid populations and virus transmission in a single application.

## 5. Conclusions

This study demonstrates the potential of RNAi-based strategies for controlling *M. persicae* through ingestion of dsRNA effectors targeting key physiological genes. We identified *ACE*, *AQP*, and *RNAHelicase* as effective individual targets, with combinations and stacked constructs further enhancing mortality. Notably, high mortality was observed even when transcript knockdown was limited, suggesting that RNAi efficacy may involve complex, multi-level disruptions beyond gene silencing alone.

By integrating individual, combinatorial, and stacked dsRNA approaches, we propose a holistic strategy for aphid control. Combined with our previous work on RNAi-mediated virus suppression, this research supports the development of a dual-action, spray-on RNAi platform capable of disrupting both insect and viral threats. Such a system offers a clean, green, and sustainable alternative to chemical pesticides for protecting crops from aphid infestation and virus transmission.

## Figures and Tables

**Figure 1 insects-16-01086-f001:**
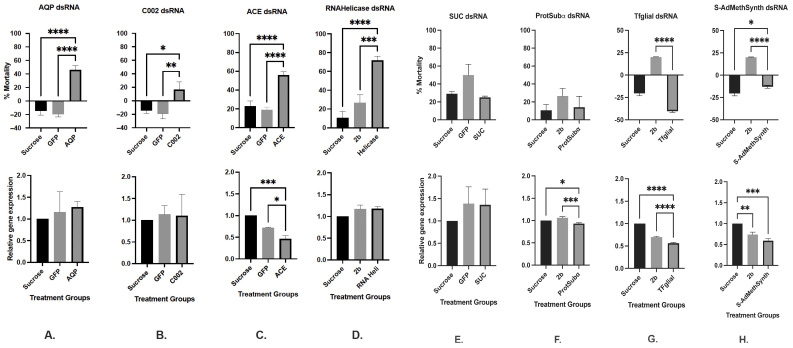
(**A**–**H**): Artificial diet individual dsRNA feeding assays. (**A**): AQP dsRNA; (**B**): C002 dsRNA; (**C**): ACE dsRNA; (**D**): RNAHelicase dsRNA. (**E**): SUC dsRNA; (**F**): ProtSubα; (**G**): Tfglial dsRNA; (**H**): S-AdMethSynth dsRNA. Mean of three (*n* = 3) independent artificial diet feeding assays showing percent mortality (Top panel) and relative gene expression (Lower panel) in *M. persicae* (n~25/dish with four replicates each treatment in each AD assay) fed on respective dsRNA effectors at 300 ng µL^−1^ each with 30% sucrose only or sucrose supplemented with 300 ng µL^−1^ of either GFP-dsRNA or CMV 2b-dsRNA non-specific controls. Levels of significance were determined by one-way ANOVA with Tukey’s multiple HSD test in GraphPad PRISM 8. Significantly different means are denoted using asterisk (* *p* < 0.05, ** *p* < 0.01, *** *p* < 0.005, and **** *p* < 0.0001).

**Figure 2 insects-16-01086-f002:**
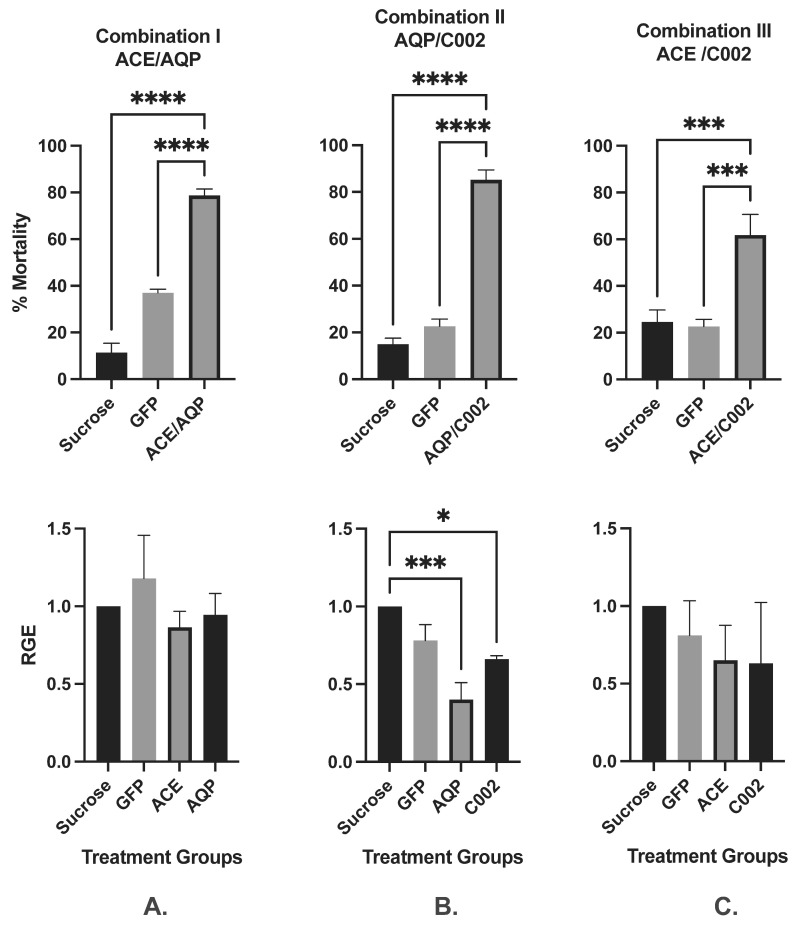
Artificial diet combined dsRNA feeding assays. (**A**): Combination I ACE/AQP dsRNA; (**B**): Combination II AQP/C002 dsRNA; (**C**): Combination III ACE/C002 dsRNA. Mean of three (*n* = 3) independent artificial diet feeding assays showing per cent mortality (Top panel) and relative gene expression (Lower panel) of *M. persicae* (n~25/dish with four replicates each treatment in each AD assay) fed on combinations of dsRNA effectors at 150 ng µL^−1^ each with 30% sucrose (30%) or non-specific GFP-dsRNA (300 ng µL^−1^) controls. Levels of significance were determined by one-way ANOVA with Tukey’s multiple HSD test in GraphPad PRISM 8. Significantly different means are denoted using asterisk (* *p* < 0.05, *** *p* < 0.005, and **** *p* < 0.0001). (RGE—Relative Gene Expression).

**Figure 3 insects-16-01086-f003:**
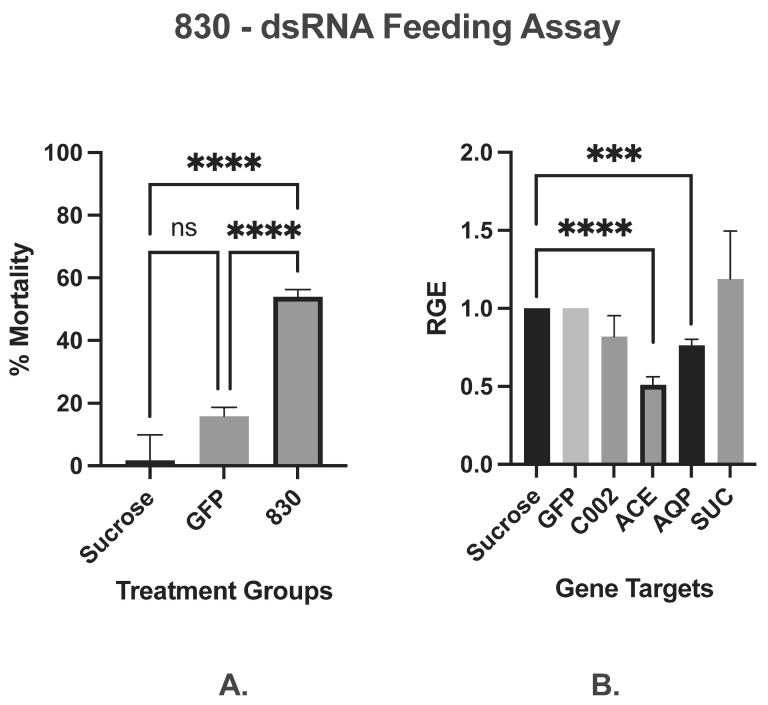
Artificial diet ‘stacked’ 830-dsRNA feeding assays. Mean of three (*n* = 3) independent artificial diet feeding assays showing per cent mortality (**A**) and relative gene expression (**B**) of C002, ACE, AQP and SUC *M. persicae* (n~25/dish with four replicates each treatment in each AD assay) fed on 830-dsRNA effectors at 300 ng µL^−1^ with 30% sucrose (30%) or non-specific GFP-dsRNA (300 ng µL^−1^) control. Levels of significance were determined by one-way ANOVA using Tukey’s multiple HSD test in GraphPad PRISM 8. Significantly different means are denoted using asterisk (*** *p* < 0.005, and **** *p* < 0.0001). (RGE—Relative Gene Expression).

**Table 1 insects-16-01086-t001:** *Myzus persicae* Gene targets, mortality and knockdown induced in artificial diet assay.

Target Gene	Name	Function	Position	Length (bp)	Sequence ID ^a^	% Mortality in AD ^b^	% Relative Gene Expression ^c^
Aquaporin	AQP	Osmoregulatory	331–830	500	KR047100	45.8% ^d^	+24%
Sucrose transporter	SUC	Osmoregulatory	1334–1683	350	KR047101	25%	+36%
Acetylcholinesterase	ACE	Neuronal functioning	1600–1999	400	KJ561353	56%	−53.7%
Salivary gland	C002	Feeding/Probing	391–890	500	EC389531.1	16.7% ^d^	+14%
Transcription factor glial cells missing	Tfglial	Neuronal functioning	588–853	266	XM 022324935	−4.38%	−44%
S-adenosylmethionine synthase	S-AdMethSynth	Nucleic acid metabolism	622–871	250	XM 022314690	−13.02%	−41%
Proteasome subunit alpha type	ProtSubAlpha	Protein metabolism	260–569	310	XM 022311085	13.84%	−6.3%
ATP-dependent RNA helicase	RNAHelicase	Nucleic acid metabolism	667–994	328	XM 022316612	71.76%	−18.3%
Stacked (AQP, C002, ACE, SUC)	830	Osmoregulatory, Neuronal function, Feeding/Probing		830 (166 bp each)		54%	−24%, −17.7%, −49%, +19.3%
Combination I (AQP/ACE)	Combination I	AQP/ACE	As Above	As Above	As Above	78.7%	−5.5%, −13.5%
Combination II (AQP/C002)	Combination II	AQP/C002	“	“	“	85.2%	−59.7%, −33.7%
Combination III (ACE/C002)	Combination III	ACE/C002	“	“	“	79.4%	−35.3%, −36.7%

^a^ NCBI accession number for *M. persicae* genes. ^b^ Values represent respective percent mortality (−) before value represents increase in aphid population over the assay period. ^c^ Values represent the respective percent knockdown of targeted gene. (+) before value represents increase in aphid gene expression. ^d^ Values represent a mortality percentage at a dsRNA dosage of 250 ng/µL. “: As above.

## Data Availability

The original contributions presented in this study are included in the article/Appendix A. Further inquiries can be directed to the corresponding author.

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
