# Peer review of "Artificial Diet Assay Screening of Candidate RNAi Effectors Against Myzus persicae (Hemiptera)"

_insects, 2025, doi:10.3390/insects16111086_

Round 1

Reviewer 1 Report

Comments and Suggestions for Authors

The manuscript titled "Artificial diet assay screening of candidtate RANi effectors against Myzus persicae (Hemiptera)" is well put study with elaborate experimental evidences.

Various physiological essential genes were studied using RNAi to green peach aphids. Aphid mortality was seen in different RNAi treatments be it individual, combinations and stacked. This study is important to fetch out some very important genes which can be used to control green peach aphid infestation via RNAi.

A few minor comments which needs to be addressed:

Line 150: Most of the feeding bioassays requires starving the aphids, were these aphids starved before giving them the water, sucrose and dsRNA?

Line 159: Were the aphids counted only adults or both instars/nymphs and adults

Fig 1a, 1b, 2 and 3: Some bars in the bar graphs are missing error bars on them, do check for consistency in all graphs provided

Supplementary table 2: Mention Tm for each primers

Author Response

Reviewer 1:

The manuscript titled "Artificial diet assay screening of candidate RNAi effectors against Myzus persicae (Hemiptera)" is well put study with elaborate experimental evidences.

We would like to thank the reviewer for their comments and quick review of our presented research. We have addressed the reviewers comments fully and clarified your concerns in the respective manuscript. Thankyou.

In addition We identified an error in the presentation of one of the figures, Mortality graph 1b (h). Here the treatment group mortality (which was negative with and increase in aphid numbers was presented as 130.02%. This was an error and should have been 13.02% this has been amended in the figure.

A modification to the graph title in Figure 2 was also conducted.

Line 66 -  reference was not formatted correctly

Various physiological essential genes were studied using RNAi to green peach aphids. Aphid mortality was seen in different RNAi treatments be it individual, combinations and stacked. This study is important to fetch out some very important genes which can be used to control green peach aphid infestation via RNAi.

A few minor comments which needs to be addressed:

  • Line 150: Most of the feeding bioassays requires starving the aphids, were these aphids starved before giving them the water, sucrose and dsRNA?

Yes. Thank you for pointing out this oversight. We have amended the relevant Material and methods section. added more detail regarding Aphid harvesting since it was queried by the second reviewer as well.

For your convenience: The aphids were starved for up to 2 hours post-harvest while setting up the assay.  The aphids were harvested into the dishes and covered with parafilm. This dish was put aside and the next dish filled. This on average took 1 – 1.5 hours. Once all the dishes were filled and para-filmed the diet was applied and the second parafilm sheet applied. This was conducted in the order the aphids were placed so the first dish to have aphids had the diet applied first etc…

  • Line 159: Were the aphids counted only adults or both instars/nymphs and adults

All aphids that were alive and in the dishes were counted. This included any nymphs that were delivered during the assay. Aborted nymphs were not counted. For the assays we harvested nymph instars 3 – 4 and adults were harvested. Over the course of the assays some adults and late-stage nymphs progressed and were able to give birth to live young. These were counted in the mortality assays. This is reflected in the aphid counts giving inverse mortality graphs ie graphs going negative due to a positive increase in numbers.

  • Fig 1a, 1b, 2 and 3: Some bars in the bar graphs are missing error bars on them, do check for consistency in all graphs provided.

This was confusing for me as well. I think what the reviewer is referring to is the normalised sucrose control bars on the Relative gene expression graphs. Normalising the data removes the variation in these data sets and therefore do not reflect a range of error in that data.

Supplementary table 2: Mention Tm for each primers

Amended: We have added the Tm of the primer sets.

Reviewer 2 Report

Comments and Suggestions for Authors

Dear authors,

your manuscript entitled ““Artificial diet assay screening of candidate RNAi effectors against Myzus persicae (Hemiptera)” addresses a highly interesting topic for developing RNAi control strategies for sucking pest insects. However, I have significant concerns regarding the reported data and the design of the artificial feeding assay, i.e. a major revision is required. Please see below my concerns:

  • In your feeding assays, you are using unformulated (i.e., unprotected) dsRNA, despite existing reports documenting nuclease activity in the midgut of M. persicae (e.g. Ghodke et al., Sci Rep. 2019; 9:11898. doi: 10.1038/s41598-019-47357-4). Could you clarify how this was addressed in your study? How do you know that the dsRNA is in your case not degraded?

  • Line 149: You report to use “~25 aphids ..“ per assay treatment. Can you specify? How many aphids are you using per treatment (on day 1)? Is this number varying?

  • Line 156: a) you report a water treatment in your bio assays, but you do not provide data of that control. From our own experiences of rearing M. persicae and performing artificial feeding experiments, we know very well that the aphids do not survive a water-only treatment over 5 days. Could you please specify? b) Even a 30% sucrose artificial diet without additional nutrients is also very challenging in terms of background mortality and level of aphid fitness over 5 days.

Could you please provide your raw survival data during artificial diet feeding expressed as number of aphids as function of treatment time (5 days, tracked every day)?

  • From our experiences, the life span of M. persicae is approximately 21 (±2) days, and individuals can vary considerably in size, fitness, and reproduction rate. Since you did not stage the aphids used in your trials, the experimental conditions are not directly comparable, which is reflected in the high background mortality observed in your control groups. For instance, you report mortality in sucrose and GFP controls ranging from -20% (Fig. 1a, A, B) to +20% (Fig. 1a, C; Fig. 1b, E), resulting in a total variability of at least 40%. In addition, you do not differentiate between adults and nymphs, which further complicates the interpretation of your results, because the nymphs may be treated with artificial diet for 1, 2, 3 or 4 days. This data set does not allow for a conclusive interpretation.

  • Could you please specify if n=3 relates to technical or biological replicates? As the aphid lifespan, size and fitness during artificial diet feeding assay is very complex, it is highly recommended to increase the number of biological replicates to at least n=7.

Unfortunately, as you can see, I have fundamental concerns regarding the reported data. Based on our experience, bioassays with Myzus persicae are highly complex and require precise aphid staging to reduce background mortality, as well as an appropriate formulation strategy to protect dsRNA against nuclease-mediated degradation.

Author Response

Reviewer 1:

Dear authors, your manuscript entitled ““Artificial diet assay screening of candidate RNAi effectors against Myzus persicae (Hemiptera)” addresses a highly interesting topic for developing RNAi control strategies for sucking pest insects. However, I have significant concerns regarding the reported data and the design of the artificial feeding assay, i.e. a major  revision is required. Please see below my concerns:

We thank the reviewer for their quick review of our presented research and appreciate the  insights, thoughts and questions regarding our manuscript. We have been able address most of the respective reviewers’ concerns.

In addition to the amendments outlined below, we identified an error in the presentation of one of the figures, Mortality graph 1b (h). Here the treatment group mortality was presented as 130.02%. This was an error and should have been 13.02% this has been amended in the figure.

A modification to the graph title in Figure 2 (b) was also conducted. Removed ‘AD’

  • In your feeding assays, you are using unformulated (i.e., unprotected) dsRNA, despite existing reports documenting nuclease activity in the midgut of M. persicae (e.g. Ghodke et al., Sci Rep. 2019; 9:11898. doi: 10.1038/s41598-019-47357-4). Could you clarify how this was addressed in your study? How do you know that the dsRNA is in your case not degraded?

Thank you for your thoughtful comment and for highlighting the important consideration of dsRNA stability in the aphid gut environment.

We fully acknowledge the presence of midgut nucleases in M. persicae, as reported by Ghodke et al. (2019, Sci Rep 9:11898; https://doi.org/10.1038/s41598-019-47357-4) and others (e.g., Christiaens et al., 2014; https://doi.org/10.1016/j.peptides.2013.12.014). In this study, we used unformulated (naked) dsRNA, and we recognize that this likely resulted in partial degradation of the dsRNA during ingestion and digestion.

Despite this, we observed consistent and statistically significant increases in mortality in aphids fed target-specific dsRNA compared to controls, suggesting that sufficient intact dsRNA was present to elicit an RNAi response. While we agree that the delivery was not optimized for nuclease resistance, the biological effects observed indicate that the dsRNA retained some functional integrity.

Furthermore, we have recently conducted complementary experiments incorporating nuclease-targeting dsRNA in combination with virus-retention-targeting dsRNAs. These studies have demonstrated enhanced effects, supporting the hypothesis that mitigating nuclease activity can improve RNAi efficacy. This work, led by a co-author who joined the lab during the course of this study, is currently being prepared as a separate manuscript focused on in planta applications of combined dsRNA treatments.

We appreciate your comment, which has allowed us to better contextualize the limitations and future directions of our work, and have ammened the discussion to reflect this clarification.

  • Line 149: You report to use “~25 aphids ..“ per assay treatment. Can you specify? How many aphids are you using per treatment (on day 1)? Is this number varying?

Thank you for your comment and the opportunity to clarify this aspect of our methodology.

We have amended the Materials and Methods section to better describe the aphid harvesting process and the number of individuals used per treatment. On average, approximately 25 aphids were deposited into each assay dish at the start of the experiment. The use of the tilde (~) symbol reflects minor variation, as in some instances up to 30 aphids may have been transferred.

Aphids were harvested from detached leaves using a gentle heat-mobilisation method. Specifically, the adaxial surface of the leaf was briefly warmed on a metal desk lampshade to encourage aphid movement. Once mobile, aphids were flicked into a high-walled plastic tray. Healthy, active individuals naturally climbed the tray walls and were collected using a gentle vacuum pooter from the rim of the container. These aphids were then deposited into 25 mm cell culture dishes via a wide-bore pipette tip and tubing system.

We acknowledge that some harvest-related attrition occurred. Therefore, mortality data were normalised to the number of aphids surviving at 24 hours post-harvest, which served as the baseline for subsequent mortality assessments. This initial number varied slightly between replicates due to minor differences in harvest efficiency and handling stress. These variations are documented in the raw count data provided in Supplementary Data File 2.

We appreciate your Question as this was also queried by another reviewer and have updated the manuscript to ensure this process is more clearly described.

  • Line 156: a) you report a water treatment in your bio-assays, but you do not provide data of that control. From our own experiences of rearing M. persicae and performing artificial feeding experiments, we know very well that the aphids do not survive a water-only treatment over 5 days. Could you please specify? b) Even a 30% sucrose artificial diet without additional nutrients is also very challenging in terms of background mortality and level of aphid fitness over 5 days.

Thank you for raising this important point.

a) The water-only treatment was initially included in our early assay development as a total mortality control. As expected, aphids did not survive beyond 72 hours under these conditions, consistently resulting in 100% mortality. However, as the study progressed we removed the water-only group from subsequent experiments due to logistical constraints. We acknowledge that its inclusion in the manuscript was an oversight, and we have now removed references to the water treatment from the Materials and Methods section to avoid confusion.

b) We fully agree that a 30% sucrose-only artificial diet presents challenges for aphid survival and fitness, particularly beyond 5 days. In our assays, the exposure period was limited to 98 hours (4 days), and we considered assays valid only when mortality in the sucrose-only control remained below 30% at the endpoint. This threshold was used to ensure that the data reflected treatment effects rather than excessive background mortality.

We appreciate your comments, which have helped us clarify these aspects of our methodology, and have revised manuscript accordingly to improve transparency.

  • Could you please provide your raw survival data during artificial diet feeding expressed as number of aphids as function of treatment time (5 days, tracked every day)?

Provided: An excel file has been submitted as a supplementary for the genes tested. I regret that the data is incomplete regarding ‘everyday counts’ as some assays were run over weekends and one can appreciate students and RA’s are a little busy on weekends.

  • From our experiences, the life span of M. persicae is approximately 21 (±2) days, and individuals can vary considerably in size, fitness, and reproduction rate. Since you did not stage the aphids used in your trials, the experimental conditions are not directly comparable, which is reflected in the high background mortality observed in your control groups. For instance, you report mortality in sucrose and GFP controls ranging from -20% (Fig. 1a, A, B) to +20% (Fig. 1a, C; Fig. 1b, E), resulting in a total variability of at least 40%. In addition, you do not differentiate between adults and nymphs, which further complicates the interpretation of your results, because the nymphs may be treated with artificial diet for 1, 2, 3 or 4 days. This data set does not allow for a conclusive interpretation.

Thank you for your detailed observations and for highlighting the importance of aphid staging in artificial diet assays. We appreciate the opportunity to clarify our experimental design and the rationale behind our approach.

We acknowledge that synchronizing aphids to a specific instar can reduce variability and improve comparability across treatments. However, in this study, our intention was to simulate a more realistic scenario reflective of protected cropping systems, where aphid populations are inherently heterogeneous in age, morphology, and reproductive status. As such, we deliberately chose not to stage aphids to a particular instar, aiming instead to assess the efficacy of dsRNA treatments under conditions that more closely resemble those encountered in production environments.

To mitigate the variability introduced by this approach, we implemented a form of population-level cohort staging. Specifically, we maintained three staggered aphid colonies and selected individuals from colonies that were 5–7 days post-infestation—representing a moderate infestation stage. At this point, colonies were robust, plants remained healthy, and a representative mix of developmental stages (primarily N3 to wingless adults, likely viviparous) was present. This strategy allowed us to avoid early-stage colonies (which were too fragile for large-scale aphid removal) and late-stage colonies (which exhibited stress indicators such as increased alate formation and fungal growth due to honeydew accumulation).

While we agree that this approach introduces some variability, we believe it provides a meaningful representation of field-relevant conditions. The observed background mortality in control groups likely reflects this natural heterogeneity, but we contend that the overall trends—particularly the increased mortality associated with target-specific dsRNA ingestion—remain biologically relevant and indicative of an RNAi effect.

We appreciate your comments, which have helped us better articulate the rationale behind our methodology, and we will revise the manuscript to make these details more explicit.

  • Could you please specify if n=3 relates to technical or biological replicates? As the aphid lifespan, size and fitness during artificial diet feeding assay is very complex, it is highly recommended to increase the number of biological replicates to at least n=7.

To clarify, n = 3 refers to the number of independent experimental trials conducted for each dsRNA effector. Within each trial, we included four biological replicates per treatment group, resulting in a total of 12 biological replicates per effector across the study.

We appreciate your recommendation to increase the number of biological replicates. While logistical constraints limited us to this design, we agree that expanding the number of biological replicates in future studies would strengthen the statistical power and robustness of the findings, particularly given the inherent variability in aphid physiology and behavior during artificial diet assays.

  • Unfortunately, as you can see, I have fundamental concerns regarding the reported data. Based on our experience, bioassays with Myzus persicae are highly complex and require precise aphid staging to reduce background mortality, as well as an appropriate formulation strategy to protect dsRNA against nuclease-mediated degradation.

We sincerely appreciate your thoughtful feedback and fully acknowledge the complexities involved in conducting bioassays with Myzus persicae. Your concerns regarding aphid staging and dsRNA formulation are well-founded and reflect the challenges that many researchers, including ourselves, continue to navigate in this field.

In designing our study, we opted for a pragmatic approach that aimed to reflect real-world conditions—specifically, the heterogeneous nature of aphid populations encountered in production settings. By using a mixed-age, morphologically diverse population, we sought to simulate the variability present in production environments where spray applications would be deployed.

We agree that the literature presents a wide range of outcomes regarding RNAi efficacy in aphids, as highlighted by Yoon et al. (https://doi.org/10.1016/j.ibmb.2020.103408), who demonstrated significant intra-population variability in RNAi susceptibility. In line with your suggestion, we have recently completed a complementary study investigating nuclease-dsRNA interactions and their role in virus retention in aphids, which we hope to report very shortly.

While we recognize that our current formulation does not include nuclease inhibitors or protective carriers, we believe the observed increases in mortality—particularly when target-specific dsRNA is ingested—are indicative of a biologically relevant RNAi effect. Moreover, the additive effects observed with dsRNA combinations further support this interpretation, even in the absence of optimized delivery strategies.

We are grateful for your insights, which have helped us to better contextualize our findings and limitations. We hope our clarifications address your concerns and demonstrate the rationale behind our 'application focused' experimental design.